# Rodent Models for Atherosclerosis

**DOI:** 10.3390/ijms27010378

**Published:** 2025-12-29

**Authors:** Linghong Zeng, Jingshu Chi, Meiqi Zhu, Hong Hao, Shiyin Long, Zhenguo Liu, Caiping Zhang

**Affiliations:** 1Department of Biochemistry and Molecular Biology, Hengyang Medical School, University of South China, Hengyang 421001, China; zlh70585687@163.com (L.Z.); zhumeiqi0207@163.com (M.Z.);; 2Division of Cardiovascular Medicine, Department of Internal Medicine, University of Nebraska Medical Center, Omaha, NE 68198, USA; jchi@unmc.edu (J.C.); hhao@unmc.edu (H.H.)

**Keywords:** atherosclerosis, high-fat and cholesterol diet, dyslipidemia, rodent model

## Abstract

Atherosclerosis, a leading cause of cardiovascular disease, is driven by a complex interplay of dyslipidemia, inflammation, and arterial plaque formation and progression. Animal models are indispensable to elucidate the pathogenesis and develop novel therapies. Rodent models are widely utilized due to their cost-effectiveness, reproducibility, and rapid disease progression. However, notable species differences exist in lipoprotein composition and lipid metabolism pathways. Mice and rats exhibit an HDL-dominant profile, whereas Syrian golden hamsters express cholesteryl ester transfer protein (CETP) and display a higher LDL fraction, but lower than that of humans, offering a model closer to human metabolically. Divergent CETP activity across species further complicates the translational relevance of the findings from these models for atherosclerosis and related metabolic disorders. This review systematically examines the key factors in rodent model selection and optimization, with consideration on the roles of sex and age. We focus on three commonly used and well-characterized rodent strains prone to atherosclerosis: C57BL/6J mice, Sprague-Dawley (SD) rats, Wistar rats, and golden hamsters. On *Apoe*^−/−^ or *Ldlr*^−/−^ backgrounds, male C57BL/6 mice, owing to their pronounced hypercholesterolemia and extended survival with high-fat diet, are preferentially used in late-stage plaque stability studies. In contrast, male SD or Wistar rats develop atherosclerosis slowly with limited lesion progression, while hamsters, despite their human-like lipid metabolism, exhibit substantial individual variability and lesions that typically arrest at early fatty streaks with poor reproducibility. Therefore, rats and hamsters are better suited for studies focusing on early disease mechanisms and human-mimetic lipid metabolism.

## 1. Background

Atherosclerosis, a leading cause of cardiovascular diseases, is closely linked to dyslipidemia (such as hypercholesterolemia and hypertriglyceridemia) and is characterized by increased atherogenic and/or decreased anti-atherogenic lipoprotein levels [1,2]. Elevated low-density lipoprotein cholesterol (LDL-C) is considered the primary risk factor for atherosclerosis [3]. Initially, excess lipids accumulate in macrophages that are transformed into foam cells in the arteries, forming fatty streaks in the early phase of atherosclerosis [4]. As the disease advances, plaques develop with a lipid-rich or necrotic core surrounded by a collagen-rich fibrous cap, leading to advanced atherosclerosis [5]. To dissect these complex pathogenic processes and explore potential interventions, optimal experimental models are essential. Given the low cost, easy maintenance, high reproducibility, and relatively rapid disease development, rodent models have been widely adopted in atherosclerosis research [6,7].

To model human dyslipidemia and atherosclerosis, rodents are usually fed high-fat, high-cholesterol (HFHC) diets. The atherogenic components of these diets can contribute to atherosclerosis through different mechanisms. First, certain lipid elements directly elevate the levels of atherogenic lipoproteins. For instance, casein-enriched diets raise beta- very low-density lipoprotein cholesterol (VLDL) levels, while suppressing the thyroid hormone axis, thereby promoting lipid deposition [8,9,10,11]. Second, the bile acid modulators, particularly cholic acid, enhance intestinal cholesterol absorption and inhibit bile acid synthesis, thus rapidly increasing plaque burden; however, their use remains controversial due to their associated hepatotoxicity [12,13,14,15]. Third, microbiota-derived metabolites, such as dietary choline that is converted to trimethylamine-N-oxide (TMAO), exhibit a clear positive correlation with atherosclerotic lesion sizes [16]. Finally, some reagents may interrupt endocrine functions. For example, pyrimidine analogs may induce hypothyroidism [17,18,19], while vitamin D3 combined with high-fat diet (HFD) could accelerate vascular calcification, facilitating the development of advanced atherosclerosis with calcified lesions [20]. Although pyrimidines and cholic acid are toxic at high doses, at appropriate dosing (~0.2% and ~0.5%, respectively), they do not appear to compromise animal health or study reproducibility and translational relevance.

The proportions of fat, cholesterol, cholate, pyrimidine compounds, and casein vary substantially across different dietary formulations, thus, significantly contributing to the outcomes and/or progression of different atherosclerosis models. Figure 1 illustrates the potential effects of each component of HFD on the development and progression of atherosclerosis in animal models. This review focuses on the effects of common dietary compositions on blood lipid profiles and the development and progression of atherosclerosis in rodent models, offering dietary recommendations to optimize experimental models for atherosclerosis research.

## 2. Optimizing Rodent Models for Atherosclerosis Research

### 2.1. Conserved Lipid Metabolism Pathways Between Rodents and Humans

Rodent models are the most used animal models for experimental investigation of atherosclerosis due to the advantages of these models, including (but not limited to) (1) similar lipid metabolisms, with mice and rats sharing the conserved pathways of cholesterol and triglyceride metabolisms with humans [21]; (2) genetic manipulability, with transgenic strains (e.g., *ApoE*^−/−^ and *Ldlr*^−/−^ mice) allowing for the precise study of gene contributions to the pathogenesis of atherosclerosis [22,23]; (3) rapid progression of hyperlipidemia and atherosclerosis, with HFD rapidly inducing dyslipidemia and atherosclerotic lesion formation, with visible aortic plaques within weeks in *ApoE*^−/−^ and *Ldlr*^−/−^ mice [24]; and (4) similar pathological features, with rodent models exhibiting intimal lipid deposition, aortic foam cell formation, fibrous caps, and advanced plaques resembling human atherosclerotic lesions [25].

### 2.2. Key Differences in Lipid Metabolisms Between Rodents and Humans

Although rodents are the most widely used models for atherosclerosis research, there are significant differences in lipid–metabolic pathways and associated genes and protein profiles between rodents and humans [26,27], including the following: (1) Different composition of apolipoprotein A (Apo A) in HDL particles—HDL contains approximately 60% Apo A-I and 20% Apo A-II in humans [28], whereas HDL is dominated by Apo A-I with negligible Apo A-II in mice or rats [29,30,31]. (2) Divergences of Apo C III in LDL particles—in humans, LDL particles are rich in ApoB-100 and Apo C-III, the latter of which contributes to atherogenicity by inhibiting lipoprotein lipase (LPL) [32], while in mice or rats, LDL particles barely contain Apo C-III [33]. (3) Absence of cholesteryl ester transfer protein (CETP) in mice—in humans, CETP transfers cholesteryl esters from HDL to VLDL/LDL, leading to the formation of cholesterol-rich LDL remnants that are cleared by hepatic LDLR (a classic reverse cholesterol transport, RCT) [34]. HDL-cholesteryl esters are directly cleared by hepatic scavenger receptor class B type I (SR-BI), resulting in an HDL-dominant lipoprotein profile with minimal LDL due to a lack of CETP in mice, whereas golden Syrian hamsters express functional CETP, with HDL/LDL ratios and dietary cholesterol responses closely resembling humans, making it the most suitable rodent model for human-like RCT studies [35,36]. (4) Difference in cholesterol synthesis, bile acid metabolism, and absorption—in mice or hamsters, bile acid mixture is more hydrophilic accompanied by higher basal hepatic cholesterol synthesis, and enhanced CYP7A1-mediated catabolism and ABCG5/G8-dependent excretion rapidly shunt excess cholesterol into the gut, limiting hepatic accumulation [37,38]. The absence of a gallbladder in rats results in a diluted, continuous flow of bile into the duodenum, thus impairing micellar solubilization and absorption of dietary cholesterol and further reducing the systemic cholesterol load [39].

However, the development and progression of atherosclerosis vary widely in different rodent models. Compared to other rodents, inbred mice are more readily available in large quantities for atherosclerosis research. In these models, cholesterol is mainly found in HDL, with small amounts in pro-atherosclerotic VLDL and LDL. Although wild-type (WT) mice on a normal diet rarely develop atherosclerosis [40], WT C57BL/6J mice are notably more susceptible to developing atherosclerosis when fed an HFD [41]. Under standard conditions, WT C57BL/6J mice maintain stable plasma cholesterol levels (73–78 mg/dL) between 10 and 25 weeks of age. When fed an HFHC diet (21% fat, 0.15% cholesterol), their blood cholesterol levels increase to 143–179 mg/dL over the same period [42]. In mice, plasma cholesterol levels of about 300 mg/dL are generally sufficient to initiate early atherosclerotic changes [15], while the levels approaching 1000 mg/dL may rapidly enhance lesion progression to advanced plaques characterized by fibrous caps, necrotic cores, and calcification (see Table 1).

Table 1 provides a summary of the blood lipid profiles and atherosclerotic lesion development in various established mouse models, including *Ldlr* knockout, *Apoe* knockout, AAV-PCSK9, and *Apo E*3*-Leiden mouse models, with different forms of HFD. This table highlights how genotype, dietary cholesterol content, and feeding duration collectively dictate the stages of atherosclerotic lesion development. *Apoe*^−/−^ mice are the most diet-responsive model, developing severe hypercholesterolemia and rapidly progressing lesions from foam-cell lesions to advanced plaques within 12–14 weeks, even on moderate-cholesterol diets. *Ldlr*^−/−^ mice require higher cholesterol and/or longer feeding durations to achieve a similar lesion severity, with advanced plaques typically appearing after ≥20 weeks. Wild-type C57BL/6J mice remain largely resistant, forming only minimal fatty streaks under comparable conditions, while AAV-PCSK9 mice exhibit an intermediate phenotype, in which LDL-driven hypercholesterolemia is sufficient to induce multiple plaques on the same diet.

### 2.3. Diet-Induced Atherosclerosis in Apoe^−/−^ and Ldlr^−/−^ Mouse Models

Both *Ldlr*^−/−^ and *Apoe*^−/−^ mice with C57BL/6J background are widely used in atherosclerosis research. ApoE is a key component of VLDL and HDL and facilitates LDLR-mediated lipoprotein clearance, thus, playing a crucial role in cholesterol and lipid transport in vivo [50,51]. Apolipoprotein B (ApoB) is a major structural component of chylomicrons and VLDL and is synthesized in the liver and intestine. Compared with *Ldlr*^−/−^ mice, *Apoe*^−/−^ mice develop more severe hyperlipidemia on the same diet and display significantly elevated serum ApoB48 levels. This finding indicates that apolipoprotein E accelerates the clearance of chylomicron and VLDL remnants independent of the LDL receptor [52,53,54].

Inducing atherosclerosis in mouse models typically requires a Western diet (WD) or an HFD or other modified diets. *Apoe*^−/−^ mice develop only moderate hypercholesterolemia on a standard diet [55]. In contrast, when fed a WD, the plasma cholesterol levels in these mice rise markedly to between 1085 and 4402 mg/dL. This substantial elevation in the plasma cholesterol levels is pathologically required to drive the lesion progression from early fibrous plaques, which emerge around 10 weeks, to advanced atherosclerotic lesions by 20 weeks [56]. Thus, feeding an HFHC diet to *Apoe*^−/−^ mice accelerates the establishment of an experimental atherosclerosis model with significant plaque burden in the aortic root, abdominal aorta, and thoracic aorta within just 13 weeks [57]. An HFHC diet with 42% of calories from fat can induce a hyperlipidemic state in *Ldlr*^−/−^ mice that is sufficient to form atherosclerotic lesions within 12 weeks, though these lesions typically present a mild phenotype of atherosclerosis [58].

The age of *Apoe*^−/−^ and *Ldlr*^−/−^ mice is a critical factor for the modeling of specific stages of atherosclerosis. *Apoe*^−/−^ mice are typically started on an HFD at 4–8 weeks of age to coincide with the onset of metabolic susceptibility. Initiating an HFD at this age, mice develop early atherosclerotic lesions by 8 weeks, with severe plaques emerging by approximately 16 weeks of HFD feeding [59]. For *Ldlr^−/−^* mice, an HFD is usually initiated at the age of 4–10 weeks. When starting HFD at the age of 10 weeks old, mice develop dyslipidemia and early lesions at around the age of 16 weeks old [46]. However, if the study objective is to investigate long-term disease progression, initiating an HFD as early as 4 weeks of age can result in the development of severe, advanced atherosclerotic lesions by 32 weeks of age [48].

The *Apoe*^−/−^ phenotype closely mirrors human remnant-driven hyperlipidemic conditions, such as type III hyperlipoproteinemia or familial dys-beta-lipoproteinemia. Consequently, this model is widely used to investigate the rapid initiation and progression of atherosclerosis driven by remnant lipoproteins. In contrast, *Ldlr*^−/−^ mice manifest primarily elevated plasma LDL-C due to defective LDL-receptor-mediated uptake, with their atherosclerotic lesions evolving more gradually. The resulting lipid profile and lesion distribution resemble the LDLR-deficient form of familial hypercholesterolemia [52], Thus, *Ldlr*^−/−^ mice are better suited for investigating the dynamic progression of early-to-mid-stage plaques under the conditions of isolated LDL elevation.

Although *Apoe*^−/−^ and *Ldlr*^−/−^mice are standard models of atherosclerosis, they carry key limitations for translational studies. Mice naturally lack CETP, and their hepatic apoB-100 secretion differs markedly from humans [60,61]. Moreover, mouse atherosclerotic lesions rarely develop to advanced vulnerable plaques with thin fibrous caps or spontaneous rupture, and limited calcification and intraplaque hemorrhage in contrast to the lesions in human subjects [62,63]. Of note, these mouse models typically require an HFD or a WD to induce the pathological changes; however, humans can develop substantial atherosclerotic plaques even under moderate circulating cholesterol levels [15]. These physiological and pathological differences can pose challenges in directly translating the findings from mouse models to human conditions. However, WD and HFHC diets are commonly used to induce atherosclerosis, and the cholesterol contents in these diets can be adjusted to meet specific experimental needs for optimal study outcomes. Table 1 summarizes the blood lipid levels and aortic plaque areas in various commonly used diet-induced *Apoe*^−/−^ and *Ldlr*^−/−^ mouse models for atherosclerosis.

### 2.4. Mouse Models for Atherosclerosis Using Genetic and Viral Approaches

#### 2.4.1. The AAV-*Pcsk9* Mouse Model Offers a Novel, Non-Genetically Modified Alternative for Experimental Research on Atherosclerosis

Proprotein convertase subtilisin/kexin type 9 (PCSK9) is a circulating protein that promotes LDLR degradation, thereby reducing LDL uptake and accelerating the development of atherosclerotic plaques. Therefore, PCSK9 has become a key target for the management of hyperlipidemia and atherosclerosis intervention [64,65].

When adeno-associated virus (AAV) encoding D374Y gain-of-function mutant of PCSK9 (AAV-Pcsk*9^DY^*) is introduced into WT mice, it results in the overexpression of PCSK9, leading to increased LDLR degradation. Feeding the AAV-*Pcsk9^DY^* mice with an HFD rapidly induces hyperlipidemia with elevated plasma cholesterol levels [66,67]. Compared with *Apoe* or *Ldlr* knockout mice, AVV-*PCSK9* mice do not require complex and time-consuming backcrossing. After 12 weeks of feeding with an HFD (0.75% cholesterol, reference S8492-E010, Ssniff), advanced atherosclerotic lesions can be observed in the aortic arch of AVV-PCSK9 mice [66]. When rAAV8-D377Y-mPCSK9 mice are fed with either a WD (D12079B, Research diets Inc. containing 21% fat and 0.21% cholesterol without added sodium cholate) or Paigen diet (D12336, Research diets Inc., New Brunswick, NJ, USA, containing 16% fat, 1.25% cholesterol, and 0.5% sodium cholate), persistent hypercholesterolemia develops in a PCSK9 dose-dependent manner [64].

#### 2.4.2. *APOE*3*-Leiden Mice Might Be a Suitable Model for Studies on the Progression and Regression of Atherosclerosis

The *ApoE***3-Leiden* gene is a mutated form of the *Apoe* gene associated with familial abnormal β-lipoproteinemia in humans. *APOE**3-Leiden mice express a variant ApoE3-Leiden protein that maintains the functional capacity but exhibits reduced binding affinity. This model preserves the full Apo E functionality with only the efflux pathway impaired, rendering its lipid and drug response profiles highly analogous to humans. Consequently, the *APOE**3-Leiden mouse model is valuable for experimental investigations of atherosclerosis [68], particularly when preservation of Apo E function and assessment of clinical lipid-lowering drug efficacy are required.

When fed an HFD, the transgenic mice overexpressing the human dysfunctional APOE*3-Leiden variant exhibits an exacerbated susceptibility of hyperlipidemia and atherosclerosis [69]. Given that ApoE3-Leiden mediates LDL/VLDL cholesterol transport in a manner similar to humans, this model offers a superior translational relevance for lipid metabolism and preclinical studies compared to *Apoe*^−/−^ mice, despite its reduced atherogenic propensity [70,71]. It has been reported that, when fed an HFHC diet for 14 weeks, female *APOE*3*-Leiden mice develop a significant level of atherosclerotic lesions. A subsequent resumption of a standard normal diet for 4 weeks, the blood cholesterol levels return to the same levels as the control *APOE**3-Leiden mice on a standard normal mouse diet, accompanied by a reduction in inflammatory cell infiltration in the atherosclerotic lesions, although no regression in atherosclerotic plaque size is observed [72], indicating that *APOE*3*-Leiden mice might be a suitable model for studies on the progression and regression of atherosclerosis. However, the *APOE***3*-Leiden mouse model exhibits significant sex-based differences, characterized by substantially higher cholesterol levels in females than in males, and necessitates a prolonged induction period for the development of atherosclerotic lesions [73].

## 3. Rat Models for Atherosclerosis

Atherosclerosis in humans progresses slowly, making it difficult to observe the early-stage disease in mouse models. Rats are naturally resistant to developing atherosclerosis and slow in lesion progression compared to mice, making them particularly suitable for studying early atherogenic processes [74]. An established method to induce atherosclerosis in rats has been reported by feeding the rats an HFD supplemented with sodium cholate, cholesterol, thiouracil, and vitamin D3 [75,76]. Among the rat models, Sprague-Dawley (SD) and Wistar rats are frequently used, though their utility depends on the specific research focus. Wistar rats display a greater susceptibility to HFD-induced metabolic abnormalities, exhibiting an earlier onset of dyslipidemia and insulin resistance compared to SD rats. Therefore, if the research focus is to study the association between metabolic syndrome and atherosclerosis, Wistar rats should be prioritized. Conversely, SD rats, while developing milder metabolic syndromes, are more susceptible to vascular changes. If the aim is to investigate the dynamic process of early atherosclerotic lesions and foam cell formation, SD rats are considered more suitable.

Following 17 weeks of HFD feeding, Wistar rats exhibit significantly elevated plasma lipid profiles (total cholesterol: 105 ± 8.6 mg/dL; triglycerides: 185.4 ± 14.9 mg/dL) compared to SD rats (total cholesterol: 80.0 ± 4.0 mg/dL; triglycerides: 40.7 ± 3.1 mg/dL) [77]. However, it is notable that while cholesterol-enriched diets induce remarkable hypercholesterolemia in Wistar rats, their triglyceride levels often remain low or decrease and even without apparent atherosclerotic plaque formation [78]. In contrast, SD rats develop hyperlipidemia by the fourth week of an HFD, with the plasma total cholesterol and triglyceride concentrations reaching 8.66 ± 1.35 mmol/L and 0.71 ± 0.46 mmol/L, respectively, accompanied by significant lipid accumulation and foam cell formation in the aortic arch [79].

Finally, like mouse models, age and HFD duration are also important factors in establishing optimal rat models of atherosclerosis. While the optimal HFD initiation window targets young adulthood (6–10 weeks) for both strains, their subsequent susceptibility to lesion development diverges significantly. Wistar rats, despite developing severe dyslipidemia, often fail to develop significant atherosclerotic lesions even after 14 weeks on an HFD [78]. In contrast, SD rats at the age of 8 to 10 weeks usually produce a more robust model, exhibiting both significant dyslipidemia and early atherosclerotic lesions by the age of about 16 weeks [80]. In addition, when a study requires a model of advanced disease, using 8-week-old SD rats and extending the HFD feeding to approximately 25 weeks are expected to generate advanced atherosclerotic plaques [81].

Given the fact that it takes longer for rats to develop atherosclerosis compared to mice, rat models are thus considered particularly valuable for studying early atherosclerosis. Table 2 summarizes the blood lipid profiles and early atherosclerosis characteristics in rat models with different HFD conditions.

## 4. Hamster Models for Atherosclerosis and Lipid Disorders

Atherosclerosis can also be induced in hamsters [86]. Unlike mice and rats, hamsters naturally express CETP, a key regulator of HDL metabolism [87]. Additionally, hamsters express Apo B100 in liver tissue, thus facilitating LDL-C uptake and degradation via the LDLR-mediated pathway [88]. The pathways for lipid metabolism in hamsters more closely resemble that of humans compared to mice [89,90]. Hamsters fed an HFHC diet for 6 weeks can be effectively induced to develop sustained hyperlipidemia and subsequent atherosclerosis [91]. Moreover, golden hamsters fed a diet containing 20% fat, 40% fructose, and 0.25% cholesterol for 2 weeks develop significant hyperlipidemia [92], and golden Syrian hamsters fed a diet (coconut oil 150 g/kg and cholesterol 30 g/kg) for 12 weeks develop visible atherosclerotic lesions in the ascending aorta and aortic arch, with plasma total cholesterol levels reaching 19.47 mmol/L and triglyceride levels 3.47 mmol/L [93]. Thus, hamsters can serve as a valuable model for studies on the regulation of lipid metabolism and the development and progression of atherosclerosis.

The enzyme lecithin-cholesterol acyltransferase (LCAT) catalyzes the esterification of free cholesterol, facilitating the maturation of HDL in blood [94]. LCAT deficiency in hamsters leads to dyslipidemia, characterized by hypertriglyceridemia, reduced HDL-C levels, and elevated TG levels, which is similar to the pathological condition of human LCAT deficiency [91]. Moreover, an HFD exacerbates dyslipidemia and accelerates the progression of atherosclerotic lesions in LCAT-deficient hamsters, making it a valuable model for studies on lipid metabolism disorders and related cardiovascular diseases, especially atherosclerosis [95].

It has been shown that heterozygous LDLR-deficient (*Ldlr^+^*^/*−*^) hamsters exhibit significant dyslipidemia similar to that observed in human subjects with familial hypercholesterolemia (FH) [96], thus serving as an ideal animal model for experimental investigations of FH and related conditions such as atherosclerosis-related coronary heart disease [97,98]. On an HFHC diet, *Ldlr^+^*^/*−*^ hamsters exhibit significantly higher levels of plasma TC and LDL-C compared to WT hamsters, while plasma TG levels remain unchanged between *Ldlr^+^*^/*−*^ and WT hamsters [99]. In homozygous LDLR-deficient (*Ldlr*^−/−^) hamsters on a normal diet, the plasma cholesterol levels could reach over 600 mg/dL and develop more severe atherosclerotic lesions than *Ldlr*^−/−^ mice and rats on a normal diet, while a 12-week HFHC diet results in severe hypercholesterolemia in *Ldlr*^−/−^ hamsters (4997 ± 233 mg/dL), significantly higher than in heterozygous (*Ldlr^+^*^/*−*^) hamsters (2081 ± 161 mg/dL). These homozygous hamsters also display more severe aortic atherosclerosis with lesion distributions similar to humans, making it a valuable model for studies on hypercholesterolemia and associated cardiovascular complications such as atherosclerosis [97].

The *Ldlr*^−/−^ hamster model offers significant advantages for studies on lipid metabolism and atherosclerosis, particularly mimicking human lipid metabolism and atherosclerotic lesion morphology. When fed a 0.5% cholesterol and 15% lard (*w*/*w*) diet for 40 days, these hamsters develop early atherosclerotic lesions in the aortic root, accompanied by markedly elevated plasma cholesterol and triglyceride levels (exceeding 3000 mg/dL), while normal hamsters show no significant lesions. Importantly, the resulting atherosclerotic lesions in hamsters closely resemble those observed in humans, with lesions distributed in the aortic arch, thoracic aorta, and abdominal aorta. This distribution pattern is consistent with the distribution seen in patients with FH [90,97].

Hamsters at the age of 8–12 weeks are usually used to establish atherosclerosis models. This age window is selected to ensure metabolic plasticity for a rapid induction of significant dyslipidemia while avoiding age-related physiological complications. Typically, when a study is initiated with 8-week-old golden hamsters, an HFD reliably induces both significant dyslipidemia and early atherosclerotic lesions within approximately 18 weeks [100]. Table 3 provides an overview of commonly used HFHC diets and the potential applications of gene editing in hamster models for atherosclerosis.

However, the use of hamsters in research has limitations. Unlike mice and rats, hamsters are not inbred, potentially leading to considerable intra-group variability. Thus, studies with mall animal numbers may face challenges in generating reproducible data. Additionally, their short tails could compromise conventional tail vein injections, a widely used approach for drug deliveries or other treatments in rodent models, potentially limiting their applicability in pharmacological, toxicological, and pathophysiological research [87,97]. Although the hamster model shows significant similarities to humans in terms of lipid metabolism and pathological features of atherosclerosis, the large individual variations and limited data reproducibility due to its non-inbred genetic background make it unsuitable for certain experiments such as small pilot studies. Therefore, when selecting an experimental model, one needs to comprehensively consider its advantages and limitations and carefully assess its applicability of each potential model to fit a specific study.

## 5. Sex Differences in Rodent Atherosclerotic Models and Potential Mechanisms

HFD may triggers different responses in male and female mice in many perspectives associated with atherosclerosis, including lipid profiles, inflammation, and endothelial injury [104,105,106]. Thus, female animals usually develop less atherosclerotic lesions than males in rodent models on an HFD [107,108].

Estrogen alleviates endothelial injury through multiple mechanisms such as regulation of glycemic homeostasis and enhancing lipid uptake and utilization [109,110,111]. C57BL/6J male mice on an identical HFD consistently exhibit higher circulating lipid levels and higher atherosclerotic burden than females. After 9 weeks on a lard-palm-oil-based diet (containing palm oil 50 g/kg, lard 50 g/kg, and cholesterol 0.4 g/kg), both male and female *Apoe*^−/−^ mice develop significant hyperlipidemia. However, the males exhibit significantly higher plasma lipid levels, and a bigger area of advanced aortic lesions compared to female mice [112]. Similarly, after 20 weeks on a WD (containing 0.21% cholesterol wt/wt, 21% fat), male *Ldlr*^−/−^ mice display markedly elevated plasma cholesterol levels compared to the females (1966 ± 412 mg/dL vs. 1171 ± 228 mg/dL), along with higher levels of atherosclerotic lesion burden, covering 13.1 ± 3.83% of the aortic surface area compared to 9.46 ± 2.75% in females [113].

Mice reach their sexual maturity early, while rats and hamsters achieve their sexual maturity late [114,115,116]. Thus, mouse models are better suited for studies to investigate the effects of sex differences on predominantly HFD-induced atherosclerosis to reduce the study duration and cost. To avoid the effects of sex hormones on metabolisms especially in lipids, oxidative stress, inflammation, and endothelial function, male animals are usually selected for studies on atherosclerosis.

Table 4 lists the time frames of atherosclerosis development and changes in blood lipid levels in male and female mice, rats, and hamsters under various HFD conditions. Compared with females, male rodents lacking the lipid-lowering and endothelial-protective estrogen consistently exhibit higher plasma cholesterol and triglyceride levels, along with more extensive atherosclerotic lesions. Although high-fat feeding classically results in more severe atherosclerosis in male than in female rodents due to estrogen-mediated protection, the development of atherosclerosis is modulated by multiple sex-specific factors, including immune responses, lipid metabolism, and genetic background. Therefore, a more severe phenotype in males is not always an inevitable outcome across all rodent models or conditions [109,117,118].

## 6. Conclusions

Atherosclerosis is a very complex pathological condition, and animal models play a critical role in defining the pathogenesis of atherosclerosis and identifying effective preventive and/or therapeutic strategies. Selection and/or optimization of suitable animal models are important for appropriate data interpretations and optimal study outcomes.

From a biological perspective, the inherent physiological traits of the species are critical factors for their suitability for experimental atherosclerosis research. Mouse, rat, and hamster are the most widely used rodent models, each with distinct advantages and limitations. Mice and rats exhibit a natural resistance to atherosclerosis, but this resistance can be effectively modified by genetic engineering, enabling them to be standard models for experimental investigation of atherosclerosis. While WT hamsters typically yield inconsistent aortic lesions with less desirable reproducibility, the model can be substantially improved via genetic modification. Such edited hamster models develop highly uniform lesions that closely resemble human atherosclerosis, thereby offering distinct advantages for translational research of atherosclerosis [125].

In addition to animal species and their genetic background, careful dietary selections play a central role in optimizing rodent models for atherosclerosis development and progression. Due to inherent limitations for each model, a combination of two or more rodent models with and without an HFD may be needed to achieve the desirable outcomes for experimental studies on atherosclerosis [92]. Male rodents are typically preferred in experimental studies, as estrogen in females has been shown to exert a protective effect against atherosclerosis [126]. Manifestations of aging generally become evident after middle age (10–14 months) [127]. To minimize the impact of aging on the development and progression of atherosclerosis, young rodents between 1 and 3 months of age are commonly used for experimental studies [128].

Details of the literature screening and exclusion process are provided in Appendix A.

## Figures and Tables

**Figure 1 ijms-27-00378-f001:**
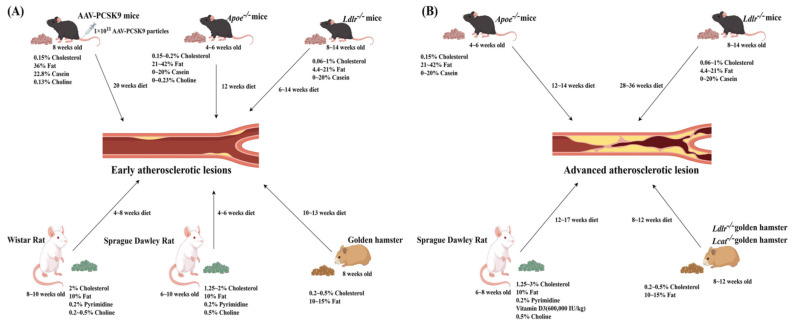
Dietary components and high-fat diet duration in the induction of early and advanced atherosclerotic lesions across animal models. Key dietary components in a high-fat diet (HFD), including cholesterol, fat, bile acid salts, choline, casein, vitamin D3, pyrimidine compounds, and thyroid hormones, have been demonstrated to promote the development of atherosclerosis. The duration of HFD feeding is a critical determinant for the stages of lesion development in experimental models. Short-term HFD primarily induces subendothelial lipid deposition and monocyte-derived macrophage infiltration, leading to early fatty streaks. In the AAV-PCSK9 model, due to the relatively modest rise in LDL-C level, a longer period is required to produce comparable early lesions. An extending HFD of ≥12 weeks progressively promotes cholesterol crystallization, phenotypic switching of vascular smooth muscle cells, and fibrous cap maturation, driving plaques into an advanced stage characterized by large necrotic cores, spotty calcification, and extensive neovascularization. (**A**) Development of early stages of atherosclerotic lesions: 4~6 weeks of age *Apoe*^−/−^ mice on an HFD for 12 weeks; 8~14 weeks of age *Ldlr*^−/−^ mice on an HFD for 6–14 weeks; 8 weeks of age AAV-PCSK9 mice on an HFD for 20 weeks; 8~10 weeks of age Wistar rat on an HFD for 8 weeks; 6–10 weeks of age Sprague-Dawley rat on an HFD for 6 weeks; 8 weeks of age golden hamster on an HFD for 10~13 weeks. (**B**) Development of advanced stages of atherosclerotic lesions: 4~6 weeks of age *Apoe*^−/−^ mice on an HFD for 12–14 weeks; 8~14 weeks of age *Ldlr*^−/−^ mice on an HFD for 28~36 weeks; 6~8 weeks of age Sprague-Dawley rat on an HFD for 17 weeks; 8~12 weeks of age *Ldlr*^−/−^ and *Lcat*^−/−^ golden hamsters on an HFD for 10~12 weeks.

**Table 1 ijms-27-00378-t001:** High-fat diet-induced atherosclerosis in mouse models: diet composition and plasma lipid profiles.

Mouse Models	Age (Weeks)	Key Components of HFD	HFD: Weeks	Fasting Plasma Cholesterol Levels	Fasting Plasma Triglyceride Levels	Stages of Atherosclerosis	References
*Apoe*^−/−^ mice	6	Western diet (WD) with standard cholesterol 0.2%, 46.1% fat, casein-vitamin tested 23.3%, and 0.23% choline tartrate	12	TC, 420 ± 14 mg/Dl HDL-C, 48 ± 2 mg/dL LDL-C, 328 ± 14 mg/dL.	133 ± 6 mg/dL	early-stage atherosclerosis, accompanied by a marked upregulation of inflammatory mediators and the development of hyperlipidemia.	[43]
*Apoe*^−/−^ mice	4	21% fat, 0.15% cholesterol, 19.5% casein, no cholate	14	TC, 865 ± 175 mg/dL	218 ± 99 mg/dL	An advanced atherosclerosis model was established with the lesion area of the ascending aorta (1.69 ± 0.23%).	[44]
*Apoe*^−/−^ mice	6	42% fat, 0.15% cholesterol and 19.5% casein (without sodium cholate)	12	TC, 1032.0 ± 50.2 mg/dL	71.6 ± 7.0 mg/dL	Advanced atherosclerotic lesions were observed, characterized by extensive and severe plaques within the aorta.	[45]
*Ldlr*^−/−^ mice	10	21% milk fat, 17% casein, 0.21% cholesterol	6	TC, 998 ± 72 mg/dL HDL-C, 83.7 ± 8.5 mg/dL.	902 ± 93 mg/dL	Atherosclerotic lesions observed in the aorta, including plaque areas of 1.29 ± 0.20% in the aortic arch and 19.36 ± 2.46% in the aortic root.	[46]
*Ldlr*^−/−^ mice	14	1% cholesterol, 4.4% fat casein-free	14	TC, 800 mg/dL	175 mg/dL	There is no obvious atherosclerotic lesion in the aorta.	[47]
			28			Severe atherosclerotic lesions were observed in the aorta, with the aortic cross-sectional lesion area measuring 337 ± 48 × 10^3^ μm^2^.	
*Ldlr*^−/−^ mice	14	21% milk fat, 0.06% cholesterol	14	TC, 1600 mg/dL	600 mg/dL	There is no obvious atherosclerotic lesion in the aorta.	[47]
			28			Severe atherosclerotic lesions characterized by fibrous caps and necrotic cores were identified in the aorta, with the aortic cross-sectional lesion area (547 ± 39 × 10^3^ μm^2^).	
*Ldlr*^−/−^ mice	8	21% fat, 0.5% cholesterol, 20% protein, 50% carbohydrates, no cholate	12	TC, 1547 mg/dL HDL, 19–39 mg/dL LDL, 387–580 mg/dL.	708.56 mg/dL	The formation of atherosclerotic plaques and lipid deposits in the aorta.	[48]
			36	TC, 1160 mg/dL HDL, 77 mg/dL LDL, 387 and 580 mg/dL	decreased	An increased area of atherosclerotic plaques and expansion of the necrotic core.	
C57BL/6J mice	8	0.15% added cholesterol (HFD-C, fat: 60% kcal; carbohydrate: 26% kcal, Bio-Serv, F3282)	20	TC, 361 ± 20 mg/dL	125 ± 9 mg/dL	A few atherosclerotic plaques were observed in the aorta.	[49]
The AAV-PCSK9 mice	8	0.15% added cholesterol (HFD-C; fat: 60% kcal; carbohydrate: 26% kcal, Bio-Serv, F3282)	20	TC, 500 ± 56 mg/dL	122 ± 19 mg/dL	Many atherosclerotic plaques were observed in the aorta.	[49]

**Table 2 ijms-27-00378-t002:** Diet composition, plasma lipid profiles, and the progression of induced atherosclerosis in rat models.

**Rat Models**	**Age (Weeks)**	**Key Components of HFD**	**HFD: X Weeks**	**Fasting Plasma** **Cholesterol Levels**	**Fasting Plasma** **Triglyceride Levels**	**Stages of** **Atherosclerosis**	**References**
Wistar Rat	8	2% cholesterol, 0.5% cholic acid, 10% lard, 5% sucrose, 0.2% 6-methyl2-thiouracil and 82.3% conventional rat food	6	TC levels, 358 ± 26 mg/dL HDL-C levels, 27 ± 7 mg/dL	32 ± 5 mg/dL.	No apparent atherosclerotic lesions were observed.	[78]
Wistar Rat	8	77.6% carbohydrate, 10% fat, 10% protein, 2% cholesterol, 0.2% bile salt and 0.2% methylthiouracil	4	TC levels, 154.66~231.99 mg/dL LDL levels, 77.33~115.99 mg/dL HDL levels, 23.2~15.5 mg/dL	86.11~129.16 mg/dL.	The hyperlipidemia model was established.	[82]
Wistar Rat	8~10	2% cholesterol HCD diet	8	TC levels, 135.33~154.66 mg/dL LDL levels, 135.33~154.66 mg/dL HDL levels, approximately 19.33 mg/dL	Approximately 43.05 mg/dL	Lipid streaks and intimal thickening were observed in the abdominal aorta, indicating the onset of early-stage atherosclerosis.	[83]
SD Rat	8	Vitamin D3, HFD with 3% cholesterol, 0.5% sodium cholate, 0.2% propylthiouracil, 5% sugar, 10% lard and 81.3% base feed (600,000 IU/kg)	17	TC levels, 102.85 ± 11.21 mg/dL LDL-C, 57.99 ± 4.25 mg/dL HDL-C, 51.04 ± 3.09 mg/dL	55.97 ± 7.75 mg/dL	Aortic lipid deposition, medial thickening, vascular calcification, and intimal rupture.	[81]
SD Rat	8~10	100 g cholesterol, 30 g propylthiouracil, and 100 g cholic acid in 1 L of peanut oil prepared fresh	6	TC levels, 202.50 ± 14.23 mg/dL LDL, 119.5 ± 11.16 mg/dL HDL-C, 31.83 ± 2.78 mg/dL	234.66 ± 10.70 mg/dL	2 rats developed lipid streaks in the aorta, while all 6 rats exhibited early atherosclerotic lesions characterized by intimal thickening and lipid droplet accumulation.	[80]
SPF adult male SD rat	Adult	2% cholesterol, 0.5% sodium cholate, 3% lard soil, 0.2% Propylthiouracil 5% refined sugar, and 89.3% base feed	4	TC, 334.66 ± 52.19 mg/dL LDL levels, 89.22 ± 39.44 mg/dL HDL-C, 32.09 ± 9.28 mg/dL	61.14 ± 39.61 mg/dL	Established an early atherosclerotic disease model, characterized by the appearance of numerous foam cells in the aorta.	[79]
SD Rat	Not specifically mentioned	81.3% basal diet, 3.5% cholesterol, 10% lard, 5.0% sucrose and 0.2% propylthiouracil.	12	TC, 231.99~579.98 mg/dL LDL-C, 115.995~579.975 mg/dL		Intimal thickening of the blood vessels was observed.	[84]
SD Rat	6	1.25% cholesterol, 0.5% sodium cholate	17	TC, 38.67.67~96.66 mg/dL LDL, 7.73~23.199 mg/dL HDL, 7.73~15.47 mg/dL.	86.11~172.22 mg/dL	Prominent atherosclerotic plaques were observed in the aorta.	[85]

**Table 3 ijms-27-00378-t003:** Diet composition, plasma lipid profiles, and induced atherosclerosis in hamster models.

**Hamster Models**	**Age (Weeks)**	**Key** **Components of HFD**	**HFD: X Weeks**	**Fasting Plasma** **Cholesterol Level**	**Fasting Plasma Triglyceride** **Levels**	**Stages of** **Atherosclerosis**	**References**
Golden hamster	8 weeks	Food with 0.2% cholesterol and 10% coconut oil	10	TC, 492 ± 67 mg/dL LDL-C, 433 ± 67 mg/dL HDL, 51 ± 6 mg/dL	333 ± 35 mg/dL.	In an early atherosclerosis model, the appearance of foam cells in the aorta.	[100]
Golden hamster	Newly weaned	200 g/kg casein, 3 g/kg methionine, 393 g/kg corn starch, 33 g/kg maltodextrin 10, 154 g/kg sucrose, 50 g/kg cellulose, 100 g/kg hydrogenated coconut oil, 2 g/kg cholesterol, 35 g/kg mineral mix and 10 g/kg vitamin mix	13	TC, 409.859 ± 15.47 mg/dL LDL-C, 305 ± 18.56 mg/dL HDL-C, 104.395 ± 6.57 mg/dL.	198.05 ± 35.30 mg/dL	Lipid streaks appeared in the aorta, occupying approximately 6% of the aortic area.	[101]
Golden hamster	8	Standard rodent meal supplemented with 0.5% cholesterol and 10% coconut oil	12	TC, 467.85 ± 42.92 mg/dL LDL-C, 196.07 ± 21.27 mg/dL HDL-C, 146.93 ± 21.65 mg/dL	938.59 ± 139.49 mg/dL	Lipid deposits were observed in the aorta.	[102]
Golden hamster	Newly weaned	0.5% cholesterol, 15% lard	12	TC, 401.18 ± 25.13 mg/dL HDL-C, 144.44 ± 14.31 mg/dL	118.27 ± 14.64 mg/dL	Lipid streaks, indicative of early atherosclerotic lesions, were observed in the aorta.	[103]
*Ldlr*^−/−^ golden hamster	10~12	0.5% cholesterol and 15% fat	12	TC, 608 ± 18 mg/dL	Approximately 400 mg/dL.	33% atherosclerotic lesions in the aortic arch, thoracic aorta, and abdominal aorta with the aortic root lesion measuring 2.9 × 10^5^ μm^2^.	[97]
*Lcat*^−/−^ golden hamster	8~12	0.5% cholesterol, 10% fat	12	TC, > 3000 mg/dL Additionally, *Lcat*^−/−^ hamsters developed hypertriglyceridemia.	15,000 mg/dL hypertriglyceridemia was observed in *Lcat*^−/−^ hamsters.	5% to 20% atherosclerotic lesions appeared in the aorta, and the aortic root plaque area ranged from 5 to 15 × 10^4^ μm^2^.	[91]

**Table 4 ijms-27-00378-t004:** Plasma lipid profiles and induced atherosclerosis in male and female rodent models.

**Rodent Models**	**Sex**	**Age (Weeks)**	**Key Components of HFD**	**HFD: X Weeks**	**Fasting Plasma Total Cholesterol Level (TC)**	**Fasting Plasma** **Triglyceride Levels**	**Lesions of** **Atherosclerosis**	**References**
*Apoe*^−/−^ mice	Male	7	50 g palm oil and 50 g/kg lard and low cholesterol content 0.4 g/kg	9	TC, 391.14 mg/dL	85.5 mg/dL	Atherosclerotic lesions in the aortic root, approximately 150 × 10^3^ μm^2^.	[112]
	Female	7		9	TC, 8.04 mg/dL.	17.4 mg/dL	Atherosclerotic lesions in the aortic root, approximately ~100 × 10^3^ μm^2^.	
*Apoe*^−/−^ mice	Male	12	Normal diet supplemented with 3% (*w*/*w*) edible whole nuts (nut group), composed of a mixture containing 50% walnuts, 25% almonds, and 25% hazelnuts	12	TC, 541.31 mg/dL.	258.33 mg/dL.	Atherosclerotic lesions in the aortic root, approximately 64.6 × 10^3^ μm^2^.	[119]
	Female	12		12	TC, 352.02 mg/dL.	172.22 mg/dL	Atherosclerotic lesions in the aortic root, approximately 63.6 × 10^3^ μm^2^.	
*Apoe*^−/−^ mice	Male	12	Normal diet supplemented with 2% (*w*/*w*) palm oil for 12 weeks	12	TC, 661.13 mg/dL.	275.55 mg/dL	Atherosclerotic lesions in the aortic root, approximately 90.2 × 10^3^ μm^2^.	[119]
	Female	12		12	TC, 417.58 mg/dL	163.61 mg/dL	Atherosclerotic lesions in the aortic root, approximately 84.2 × 10^3^ μm^2^.	
*Ldlr*^−/−^ mice	Male	5~8	1.25% cholesterol, 6% fat, minimum essential dietary requirements of vitamin E, without cholate	12	TC, 242.7 ± 7.2 mg/dL	Unavailable	5~10% atherosclerotic lesion area in the aorta.	[120]
	Female			12	TC, 209.3 ± 3.3 mg/dL	Unavailable	~5% atherosclerotic lesion area in the aorta.	
*Ldlr*^−/−^ mice	Male	6~8	21% butterfat and 0.15% cholesterol	12	TC, 1791.5 ± 103.07 mg/dL.	490.82 ± 52.53 mg/dL	11.7% lesion area in the aorta, 12% the necrotic area.	[121]
	Female			12	TC, 1353.82 ± 93.15 mg/dL	490.82 ± 32.72 mg/dL	11.5% atherosclerotic lesion area in the aorta.	
*Ldlr*^−/−^ mice	Male	8~12	42% fat and 0.2% cholesterol	12	non-HDL-C, 1600 ~1800 mg/dL	800 ~900 mg/dL	20% atherosclerotic lesion area in the aorta.	[122]
	Female			12	non-HDL-C, 1200 ~1400 mg/dL	200 ~300 mg/dL	25% atherosclerotic lesion area in the aorta.	
*SD Rat*	Male	12	Regular diet supplemented with 2% cholesterol	12	TC, 74 ± 5 mg/dL LDL-C, 35 ± 4 mg/dL HDL, 18 ± 1 mg/dL	130 ± 12 mg/dL	Dysfunction of the endothelium in the aorta.	[123]
	Female			12	TC, 71 ± 4 mg/dL LDL-C, 17 ± 1 mg/dL HDL-C, 21 ± 1 mg/dL	168 ± 4 mg/dL	Dysfunction of the endothelium in the aorta.	
Golden hamster	Male	10	10% coconut oil and 0.05% cholesterol	12	TC, 289.60 ± 12.37 mg/dL	456.37 ± 42.2 mg/dL	The aortic fatty streak appeared in the aorta.	[124]
	Female			12	TC, 246.04 ± 8.89 mg/dL	490.5 ± 18.08 mg/dL	Lipid streaks appeared in the aorta.	

## Data Availability

The original contributions presented in this study are included in the original article cited in this review. Further inquiries can be directed to the corresponding author(s).

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
