# Peer review of "Rodent Models for Atherosclerosis"

_ijms, 2025, doi:10.3390/ijms27010378_

Round 1
Reviewer 1 Report
Comments and Suggestions for Authors
In this study, the authors present the results of an analysis of the use of animal models to study the pathogenesis of atherosclerosis in humans. The authors draw attention to the differences and limitations in the use of animal models. The current manuscript is interesting, but there are some points that need to be improved.
Major comments:
- The methods section is necessary for the review. A flowchart of the initial publications (PubMed etc.) search is needed, how many papers from this search are included in the review? How many papers were omitted? Was the search done by more than one person?
- Insufficient attention was paid to the limitations of the study related to differences in normal lipid metabolism in humans and rodents. Special attention should be paid to differences in the metabolic pathways, genes, and proteins involved in lipid metabolism in humans and rodents. Differences in the effects of diets may be related to these features. Differences in normal lipid metabolism may be crucial for studying the atherosclerotic process in humans in animal models. It is advisable to prepare an additional section in the article devoted to this issue. It is advisable to prepare a table summarizing these limitations of the method of using rodents as a model of atherosclerosis. This table should include limitations unrelated to the gender and age of the rodents.
Minor comments:
- Please check the description of the picture 1. There are differences in the numbers in the picture and in the numbers in the description under this picture. (В) ... 8–12 weeks of age Ldlr-/- and Lcat-/- golden hamsters on a HFD for 10–13 weeks.
- Please check lines 103-105 on page 4. Is it the name of a table or part of a text?
Table 1. summarizes the profiles of blood lipid levels and atherosclerotic lesion development in various established mouse models, including Ldlr knockout, Apoe knockout, AAV-PCSK9, and Apo E*3-Leiden mouse models, with different forms of HFD.
Table 1. High-Fat Diet–Induced Atherosclerosis in Mouse Models: Diet Composition, Plasma Lipid Profiles.
Author Response
Comments 1: The methods section is necessary for the review. A flowchart of the initial publications (PubMed etc.) search is needed, how many papers from this search are included in the review? How many papers were omitted? Was the search done by more than one person?
Response 1: Thank the reviewer for the very thoughtful comments. A separate “Supplementary Materials” document with inclusion of a new figure showing the literature screening and exclusion flow has been included in the present review paper.
Comments 2: Insufficient attention was paid to the limitations of the study related to differences in normal lipid metabolism in humans and rodents. Special attention should be paid to differences in the metabolic pathways, genes, and proteins involved in lipid metabolism in humans and rodents. Differences in the effects of diets may be related to these features. Differences in normal lipid metabolism may be crucial for studying the atherosclerotic process in humans in animal models. It is advisable to prepare an additional section in the article devoted to this issue. It is advisable to prepare a table summarizing these limitations of the method of using rodents as a model of atherosclerosis. This table should include limitations unrelated to the gender and age of the rodents.
Response 2: Extensive modifications have been made to adequately address these comments in the revised manuscript (on page 2, lines 49 to 64). Thank you.
Comments 3: Please check the description of the figure 1. There are differences in the numbers in the picture and in the numbers in the description under this picture. (В) ... 8–12 weeks of age Ldlr-/- and Lcat-/- golden hamsters on a HFD for 10–13 weeks.
Response 3: Sorry for the typing errors. Corrections have been made according. Thank you.
Comments 4: Please check lines 103-105 on page 4. Is it the name of a table or part of a text?
Response 4: Sorry for the confusion. Changes in layout of the manuscript has been made to clearly separate each section on page 4, lines 103-105. Thank you.

Reviewer 2 Report
Comments and Suggestions for Authors
Thank you for the opportunity to review this manuscript. The article aims to discuss the selection and optimization of rodent models for atherosclerosis research, comparing commonly used species and highlighting how genetic and dietary manipulations influence disease development. It also addresses considerations such as sex, age, and translational relevance. The topic is timely, relevant, and of interest to researchers.
TITLE
- The title is clear and informative, but you may consider removing “Selection and Optimization” if the review also intends to provide critical evaluations rather than a how-to framework; alternatively, explicitly emphasize in the introduction that these aspects are central objectives.
ABSTRACT
- The abstract provides a solid overview, but it may benefit from a more explicit statement of the main conclusions or practical recommendations derived from your comparison. Readers will better understand what new insights this review contributes.
- Consider briefly acknowledging limitations or gaps in current rodent models to frame the review’s relevance and critical perspective.
- The sentence discussing species roles (“positioning mice… rats and hamsters…”) is informative but may be perceived as a strong generalization; consider adding nuance regarding experimental variability and model-specific constraints.
BACKGROUND
- You may consider reorganizing the first paragraph to improve narrative flow—moving from human disease mechanisms to animal model relevance in a more cohesive progression.
- Several mechanistic details (e.g., foam cell formation, plaque maturation) are well summarized. Still, they could benefit from additional contextualization on how closely these features recapitulate human disease, which would strengthen the justification for model choice.
- The discussion of dietary components is informative, yet it reads as a dense enumeration. Grouping these components into functional categories (e.g., pro-atherogenic lipids, bile acid modifiers, microbiota-derived metabolites, endocrine disruptors) could improve conceptual clarity and help readers understand mechanistic distinctions.
- When discussing controversial elements such as cholate and pyrimidine compounds, consider adding brief explanations of the implications for study reproducibility and translational relevance. This would enhance the critical evaluation aspect of the review.
- The rationale for linking diet composition to specific disease stages in Figure 1 is compelling; however, the text could more explicitly articulate why specific models require longer or shorter diet durations and how these durations impact phenotype severity.
SECTION 2
- This section provides a thorough overview of the advantages of rodent models, but it may benefit from a more transparent structure that distinguishes general strengths from model-specific considerations. Currently, mechanistic explanations, comparative statements, and diet-induced effects appear interwoven, which can reduce readability.
- The introductory list of advantages is informative, though converting the numerically enumerated items (1–4) into prose or a schematic could improve narrative flow. Additionally, a more explicit acknowledgment of rodents' limitations as translational models at the outset may help balance.
- The transition from general advantages to species- and strain-specific lipid metabolism is abrupt. Consider adding a sentence to frame this shift as a discussion of variability in disease susceptibility despite shared overall advantages.
- Some data presentations (e.g., plasma cholesterol ranges, diet compositions) are highly detailed; while valuable, they could be better contextualized by briefly summarizing key implications (e.g., how specific threshold concentrations correlate with lesion severity).
- This subsection presents an excellent mechanistic comparison of Apoe⁻/⁻ and Ldlr⁻/⁻ mice; however, it would be improved by explicitly highlighting the practical implications for model selection. For example, clarify which model is better suited for early-stage vs. advanced lesion studies and which better reflects specific human lipid abnormalities.
- The discussion of ApoB mRNA editing is highly technical; consider briefly explaining why these interspecies differences are relevant for interpreting lipid profiles and plaque phenotypes. Without this linkage, the mechanistic detail may feel disconnected from the main narrative.
- Age-dependent susceptibility is well documented, but it may be helpful to include a concluding synthesis describing how age, diet type, and genetic background interact to determine lesion progression rates. This could reduce fragmentation.
- The table provides extensive quantitative detail. Consider adding a short interpretive paragraph before or after Table 1 that summarizes the major trends (e.g., which diets most rapidly induce advanced plaques, and the relative responsiveness of each model).
- The section clearly articulates limitations of classic knockout models and effectively motivates the need for alternative approaches; however, consider briefly summarizing when researchers should favor AAV-PCSK9 or ApoE3-Leiden models over traditional knockouts to enhance practical applicability.
- The description of AAV-PCSK9 models is detailed but may benefit from clarifying differences between constructs (e.g., D374Y vs. D377Y) and whether these produce distinct lipid phenotypes. This would improve interpretability for readers unfamiliar with PCSK9 variants.
- The discussion of ApoE3-Leiden mice emphasizes translational value but could also include a short critique regarding their known limitations (e.g., variability in lesion development, sex-dependent responses), which would balance the assessment.
- The age-related considerations are valuable, but the narrative could be streamlined by presenting the two models separately with clearer subheadings or explicit comparative statements.
SECTION 3
- The section effectively justifies the value of rat models for studying early atherosclerosis. Still, the opening paragraph would benefit from more precise articulation of why slow lesion progression is advantageous for mechanistic studies (e.g., greater resolution of pre-lesional events).
- The presentation of dietary formulations is thorough. Yet, the narrative is dense and could be improved by grouping diet-induced outcomes into thematic subsections (e.g., “Hyperlipidemic Responses,” “Lesion Formation,” “Strain-Specific Differences”). This would help readers navigate the large number of experimental conditions.
- The comparison between Sprague-Dawley and Wistar rats is valuable. Still, the implications for model selection should be made more explicit (e.g., when Wistar rats are preferred for metabolic syndrome models vs. when SD rats are better for generating early lesions).
- Several diets contain multiple agents (cholesterol, sodium cholate, thiouracil, vitamin D₃); however, the rationale for including each agent is not discussed. A brief explanation of cholate as an accelerator of cholesterol absorption, or of thiouracil as a thyroid inhibitor that increases lipid levels, would strengthen the mechanistic understanding.
- Age-dependent susceptibility is well described, but consider adding a short synthesis comparing optimal windows across strains to avoid redundancy.
SECTION 4
- This section clearly identifies unique features of hamster lipid metabolism (CETP expression, hepatic ApoB100 production), effectively highlighting their translational relevance. Consider consolidating this information into a dedicated “Physiological Advantages” introductory paragraph for better coherence.
- The description of multiple hamster diets is informative, but the sequence of studies feels somewhat fragmented. A chronological or thematic structure (e.g., “Diet-Induced Hyperlipidemia,” “Gene-Modified Hamster Models,” “Lesion Progression”) would improve readability.
- The section on LCAT-deficient and LDLR-deficient hamsters is strong. Still, it would benefit from a direct comparison to mouse and rat knockout models to emphasize the improved human lipid resemblance.
- When describing homozygous vs. heterozygous Ldlr mutations, it may be helpful to briefly explain phenotypic penetrance and why hamsters respond more severely than mice (e.g., due to CETP activity and VLDL/LDL distribution).
- The claim that hamster lesions “closely resemble those observed in humans” is essential; consider giving one example (e.g., fibrous caps, necrotic cores) to substantiate the statement.
- The section ends by clearly acknowledging limitations (lack of inbred strains, injection challenges), which strengthens balance. You might consider adding a concluding sentence that summarizes the trade-offs (high translational value vs. practical constraints).
TABLES 2-4
- The tables contain extensive quantitative detail, but several show inconsistencies in units (mg/dL vs mmol/L) and occasionally missing or misaligned data (e.g., ranges listed with mismatched units for HDL). These inconsistencies may compromise interpretation.
- Some diet compositions include long, unbroken ingredient lists. Consider standardizing to a “% fat / % cholesterol/additives” structure or moving detailed recipes to supplementary materials, especially when multiple tables repeat similar formulations.
- Clarify whether lipid values represent fasting or non-fasting samples, as this affects comparability across models.
- Several entries describing lesion development could be made more specific (e.g., “lipid streaks” vs. “fibrofatty lesions,” “medial thickening,” “calcification”), which would help readers assess model suitability for distinct disease stages.
- For Table 4, sex differences are an essential inclusion, but the narrative that follows the table does not fully integrate these findings. A short synthesis (e.g., males tend to show greater dyslipidemia in Apoe⁻/⁻ and Ldlr⁻/⁻ mice; females show comparable or sometimes greater lesion percentages depending on diet) would enhance cohesion.
SECTION 5
- The section provides a clear rationale for examining sex differences in rodent atherosclerosis models. However, the narrative could benefit from a more systematic structure—e.g., separating (1) lipid metabolism differences, (2) inflammation and endothelial function, and (3) hormonal contributions. This would aid readers in navigating the mechanistic complexity.
- The text appropriately highlights estrogen’s protective actions, but the description is somewhat brief. Consider expanding on specific pathways—e.g., estrogen receptor-α–mediated effects on hepatic lipid metabolism, antioxidant gene regulation, nitric oxide bioavailability, or macrophage phenotype modulation—to reinforce mechanistic depth.
- The comparative examples with Apoe-/- and Ldlr-/- mice are strong. However, integrating these findings with the preceding mechanistic discussion (i.e., linking higher male lipid levels to lower estrogen-mediated vascular protection) would produce a more cohesive argument.
- While the section notes that females typically develop fewer lesions, it would be scientifically balanced to add that the magnitude of sex differences varies by strain, diet composition, and age. Some studies have also reported cases in which females show similar lesion areas despite lower lipid levels, potentially due to differences in immune responses—this nuance would strengthen the analysis.
CONCLUSIONS
- The conclusion succinctly summarizes the central themes of the manuscript. Still, it could be strengthened by distinguishing more explicitly between (a) biological considerations and (b) practical considerations in model selection.
- The statement that mice and rats are “naturally resistant” to atherosclerosis requires slight refinement. Although wild-type rodents are resistant, common knockout strains (Apoe-/-, Ldlr-/-) are not; you might clarify that genetically modified models overcome this resistance.
- Similarly, the statement that hamster lesions are “often inconsistent and poorly reproducible” could be qualified: while some studies report variability due to lack of inbred strains, the Ldlr-/- hamster model has shown highly consistent and human-like lesions.
- The conclusion would benefit from explicitly referring back to earlier discussions on dietary components (cholesterol, cholate, casein, choline). A short sentence synthesizing how diet interacts with genetic background to determine lesion severity would give the conclusion more coherence.
- The statement that “a combination of two or more rodent models… may be needed” is strong but would be more impactful if paired with one concrete example (e.g., pairing Apoe-/- mice for early lesion formation with hamsters for human-like lipoprotein profiles).
- The discussion of sex hormones and aging is appropriate. You may also note that although young rodents are commonly used, age-related atherosclerosis (e.g., in Ldlr-/- mice) is increasingly studied for translational relevance—acknowledging this trend demonstrates awareness of diverse research aims.
Thank you once again for the opportunity to review this manuscript. I appreciate the authors’ considerable effort in compiling a comprehensive and detailed overview of rodent models for atherosclerosis research. The work demonstrates a clear commitment to scientific rigor and provides valuable guidance for investigators in the field. I hope that the comments and suggestions offered here support further refinement of the manuscript.
Author Response
Comments 1: The title is clear and informative, but you may consider removing “Selection and Optimization” if the review also intends to provide critical evaluations rather than a how-to framework; alternatively, explicitly emphasize in the introduction that these aspects are central objectives?
Response 1:Thank you for your valuable suggestion. We removed “Selection and Optimization” as suggested.
Comments 2: The abstract provides a solid overview, but it may benefit from a more explicit statement of the main conclusions or practical recommendations derived from your comparison. Readers will better understand what new insights this review contributes. Consider briefly acknowledging limitations or gaps in current rodent models to frame the review’s relevance and critical perspective.
Response 2: The abstract has been expanded as suggested in the revised manuscript. Thank you.
Comments 3: Consider briefly acknowledging limitations or gaps in current rodent models to frame the reviews relevance and critical perspective.
Response 3: Done as suggested (on page 1, lines 15 to 21). Thank you.
Comments 4: The sentence discussing species roles (“positioning mice… rats and hamsters…”) is informative but may be perceived as a strong generalization; consider adding nuance regarding experimental variability and model-specific constraints
Response 4: Wording has been modified as suggested (on page 1, lines 23 to 25).
Comments 5: You may consider reorganizing the first paragraph of background to improve narrative flow—moving from human disease mechanisms to animal model relevance in a more cohesive progression?
Response 5: Done as suggested. We appreciated the valuable comments
Comments 6: Several mechanistic details (e.g., foam cell formation, plaque maturation) are well summarized. Still, they could benefit from additional contextualization on how closely these features recapitulate human disease, which would strengthen the justification for model choice.
Response 6: We appreciate the comments. Additional materials have been added as suggested (on page 2 to 3, lines 77 to 86).
Comments 7: The discussion of dietary components is informative, yet it reads as a dense enumeration. Grouping these components into functional categories (e.g., pro-atherogenic lipids, bile acid modifiers, microbiota-derived metabolites, endocrine disruptors) could improve conceptual clarity and help readers understand mechanistic distinctions.
Response 7: Great point. The section has been reorganized as recommended (on page 2, lines 51 to 62). Thank you.
Comments 8: When discussing controversial elements such as cholate and pyrimidine compounds, consider adding brief explanations of the implications for study reproducibility and translational relevance. This would enhance the critical evaluation aspect of the review.
Response 8: Done as suggested (on page 1, lines 63 to 65). Thank you
Comments 9: The rationale for linking diet composition to specific disease stages in Figure 1 is compelling; however, the text could more explicitly articulate why specific models require longer or shorter diet durations and how these durations impact phenotype severity.
Response 9: More information has been added in the revised manuscript (on page 1, lines 92 to 99). We appreciated the constructive suggestion.
Comments 10: This section provides a thorough overview of the advantages of rodent models, but it may benefit from a more transparent structure that distinguishes general strengths from model-specific considerations. Currently, mechanistic explanations, comparative statements, and diet-induced effects appear interwoven, which can reduce readability.
Response 10: We reorganized some parts of the materials as suggested. Thank you.
Comments 11: The introductory list of advantages is informative, though converting the numerically enumerated items (1–4) into prose or a schematic could improve narrative flow. Additionally, a more explicit acknowledgment of rodents' limitations as translational models at the outset may help balance.
Response 11: The “Introduction” has been extensively revised accordingly. Thank you.
Comments 12: The transition from general advantages to species- and strain-specific lipid metabolism is abrupt. Consider adding a sentence to frame this shift as a discussion of variability in disease susceptibility despite shared overall advantages.
Response 12: Done as suggested. Thank you.
Comments 13: Some data presentations (e.g., plasma cholesterol ranges, diet compositions) are highly detailed; while valuable, they could be better contextualized by briefly summarizing key implications (e.g., how specific threshold concentrations correlate with lesion severity).
Response 13: Changes have been made accordingly (on page 4, lines 143 to 145). Thank you.
Comments 14: This subsection presents an excellent mechanistic comparison of Apoe-/- and Ldlr-/- mice; however, it would be improved by explicitly highlighting the practical implications for model selection. For example, clarify which model is better suited for early-stage vs. advanced lesion studies and which better reflects specific human lipid abnormalities.
Response 14: We addressed this concern on page 6, lines 193 to 202 in the revised manuscript. Thank you.
Comments 15: The discussion of ApoB mRNA editing is highly technical; consider briefly explaining why these interspecies differences are relevant for interpreting lipid profiles and plaque phenotypes. Without this linkage, the mechanistic detail may feel disconnected from the main narrative.
Response 15: Thank you for the comments. We have removed the materials in the revised manuscript.
Comments 16: Age-dependent susceptibility is well documented, but it may be helpful to include a concluding synthesis describing how age, diet type, and genetic background interact to determine lesion progression rates. This could reduce fragmentation.
Response 16: Done as suggested (on page 11, lines 416 to 432). Thank the reviewer for this insightful suggestion.
Comments 17: The table provides extensive quantitative detail. Consider adding a short interpretive paragraph before or after Table 1 that summarizes the major trends (e.g., which diets most rapidly induce advanced plaques, and the relative responsiveness of each model).
Response 17: Done as suggested (on page 4, lines 146 to 158). Thank you.
Comments 18: The section clearly articulates limitations of classic knockout models and effectively motivates the need for alternative approaches; however, consider briefly summarizing when researchers should favor AAV-PCSK9 or ApoE*3-Leiden models over traditional knockouts to enhance practical applicability.
Response 18: Clarifications have been made to address this concern (on page 7, lines 222 to 225 and on page 7, lines 240 to 247).
Comments 19: The description of AAV-PCSK9 models is detailed but may benefit from clarifying differences between constructions (e.g., D374Y vs. D377Y) and whether these produce distinct lipid phenotypes. This would improve interpretability for readers unfamiliar with PCSK9 variants.
Response 19: Thank you for the insightful comment. Clarifications have been made as suggested (on page 7, lines 226 to 228).
Comments 20: The discussion of ApoE*3-Leiden mice emphasizes translational value but could also include a short critique regarding their known limitations (e.g., variability in lesion development, sex-dependent responses), which would balance the assessment.
Response 20: Done as suggested (on page 8, lines 260 to 263). Thank you.
Comments 21: The age-related considerations are valuable, but the narrative could be streamlined by presenting the two models separately with clearer subheadings or explicit comparative statements.
Response 21: Done as suggested (on page 6, lines 185 to 194). We appreciate the reviewer’s constructive suggestions.
Comments 22: The section effectively justifies the value of rat models for studying early atherosclerosis. Still, the opening paragraph would benefit from more precise articulation of why slow lesion progression is advantageous for mechanistic studies (e.g., greater resolution of pre-lesional events).
Response 22: Changes have been made accordingly (on page 4 to 5, lines 108 to 131). Thank you.
Comments 23: The presentation of dietary formulations is thorough. Yet, the narrative is dense and could be improved by grouping diet-induced outcomes into thematic subsections (e.g., “Hyperlipidemic Responses,” “Lesion Formation,” “Strain-Specific Differences”). This would help readers navigate the large number of experimental conditions.
Response 23: As suggested, we have extensively reorganized the materials in this section (on page 8, lines 265 to 300). Thank you.
Comments 24: The comparison between Sprague-Dawley and Wistar rats is valuable. Still, the implications for model selection should be made more explicit (e.g., when Wistar rats are preferred for metabolic syndrome models vs. when SD rats are better for generating early lesions).
Response 24: We appreciate it. Changes have been made to address this concern (on page 8, lines 271 to 279).
Comments 25: Several diets contain multiple agents (cholesterol, sodium cholate, thiouracil, vitamin D₃); however, the rationale for including each agent is not discussed. A brief explanation of cholate as an accelerator of cholesterol absorption, or of thiouracil as a thyroid inhibitor that increases lipid levels, would strengthen the mechanistic understanding.
Response 25: Done as suggested (on page 2, lines 51 to 62).
Comments 26: Age-dependent susceptibility is well described, but consider adding a short synthesis comparing optimal windows across strains to avoid redundancy.
Response 26: Done as suggested (on page 8, lines 290 to 295).
Comments 27: This section clearly identifies unique features of hamster lipid metabolism (CETP expression, hepatic ApoB100 production), effectively highlighting their translational relevance. Consider consolidating this information into a dedicated “Physiological Advantages” introductory paragraph for better coherence.
Response 27: Done as suggested. Thank you.
Comments 28: The description of multiple hamster diets is informative, but the sequence of studies feels somewhat fragmented. A chronological or thematic structure (e.g., “Diet-Induced Hyperlipidemia,” “Gene-Modified Hamster Models,” “Lesion Progression”) would improve readability.
Response 28: Changes have been made accordingly to address this concern (on page 9 to 10, lines 305 to 372).
Comments 29: The section on LCAT-deficient and LDLR-deficient hamsters is strong. Still, it would benefit from a direct comparison to mouse and rat knockout models to emphasize the improved human lipid resemblance.
Response 29: Done as suggested (on page 9, lines 333 to 338). Thank you.
Comments 30: When describing homozygous vs. heterozygous Ldlr mutations, it may be helpful to briefly explain phenotypic penetrance and why hamsters respond more severely than mice (e.g., due to CETP activity and VLDL/LDL distribution).
Response 30: Changes have been made as suggested (on page 9, lines 306 to 309).
Comments 31: The claim that hamster lesions “closely resemble those observed in humans” is essential; consider giving one example (e.g., fibrous caps, necrotic cores) to substantiate the statement.
Response 31: Done as suggested (on page 9, lines 349 to 350). I appreciated the comment.
Comments 32: The section ends by clearly acknowledging limitations (lack of inbred strains, injection challenges), which strengthens balance. You might consider adding a concluding sentence that summarizes the trade-offs (high translational value vs. practical constraints).
Response 32: Done as suggested (on page 10, lines 366to 372).
Comments 33: The tables contain extensive quantitative detail, but several show inconsistencies in units (mg/dL vs mmol/L) and occasionally missing or misaligned data (e.g., ranges listed with mismatched units for HDL). These inconsistencies may compromise interpretation.
Response 33: Changes and corrections have been made to address this concern.
Comments 34: Some diet compositions include long, unbroken ingredient lists. Consider standardizing to a “% fat / % cholesterol/additives” structure or moving detailed recipes to supplementary materials, especially when multiple tables repeat similar formulations.
Response 34: Thank you for this thoughtful suggestion. After careful consideration, we have decided to keep the full dietary formulations in the manuscript since the exact nutrient concentrations are critical and even minor variations in trace additives can markedly influence atherosclerotic phenotypes to avoid any misunderstanding.
Comments 35: Clarify whether lipid values represent fasting or non-fasting samples, as this affects comparability across models.
Response 35: Clarifications have been made accordingly. Thank you.
Comments 36: Several entries describing lesion development could be made more specific (e.g., “lipid streaks” vs. “fibrofatty lesions,” “medial thickening,” “calcification”), which would help readers assess model suitability for distinct disease stages.
Response 36: Changes have been made to address this concern.
Comments 37: For Table 4, sex differences are an essential inclusion, but the narrative that follows the table does not fully integrate these findings. A short synthesis (e.g., males tend to show greater dyslipidemia in Apoe⁻/⁻ and Ldlr⁻/⁻ mice; females show comparable or sometimes greater lesion percentages depending on diet) would enhance cohesion.
Response 37: We made changes on page 10 to 11, lines 400 to 409 to address this concern.
Comments 38: The section provides a clear rationale for examining sex differences in rodent atherosclerosis models. However, the narrative could benefit from a more systematic structure—e.g., separating (1) lipid metabolism differences, (2) inflammation and endothelial function, and (3) hormonal contributions. This would aid readers in navigating the mechanistic complexity.
Response 38: Changes have been made to address this concern. Thank you.
Comments 39: The text appropriately highlights estrogen’s protective actions, but the description is somewhat brief. Consider expanding on specific pathways—e.g., estrogen receptor-α–mediated effects on hepatic lipid metabolism, antioxidant gene regulation, nitric oxide bioavailability, or macrophage phenotype modulation—to reinforce mechanistic depth.
Response 39: We appreciated the comments, but we felt that a detailed discussion on the effects of estrogen on atherosclerosis appears to be beyond the scope of the present review article. It could be addressed in a separate paper.
Comments 40: The comparative examples with Apoe-/- and Ldlr-/- mice are strong. However, integrating these findings with the preceding mechanistic discussion (i.e., linking higher male lipid levels to lower estrogen-mediated vascular protection) would produce a more cohesive argument.
Response 40: Thank the reviewer for the insightful comments. Again, it is felt that a detailed discussion on “estrogen-mediated effects on lipid metabolisms and vascular function” seems to be beyond the scope of the discussion of the present review article.
Comments 41: While the section notes that females typically develop fewer lesions, it would be scientifically balanced to add that the magnitude of sex differences varies by strain, diet composition, and age. Some studies have also reported cases in which females show similar lesion areas despite lower lipid levels, potentially due to differences in immune responses—this nuance would strengthen the analysis.
Response 41: Thank the reviewer for very thoughtful comments. We briefly discussed this in the revised manuscript on page 11, lines 421 to 426.
Comments 42: The conclusion succinctly summarizes the central themes of the manuscript. Still, it could be strengthened by distinguishing more explicitly between (a) biological considerations and (b) practical considerations in model selection.
Response 42: Done as suggested (on page 11, lines 413 to 432). Thanks.
Comments 43: The statement that mice and rats are “naturally resistant” to atherosclerosis requires slight refinement. Although wild-type rodents are resistant, common knockout strains (Apoe-/-, Ldlr-/-) are not; you might clarify that genetically modified models overcome this resistance.
Response 43: Done as suggested on page 16 to 17, lines 444 to 446.
Comments 44: Similarly, the statement that hamster lesions are “often inconsistent and poorly reproducible” could be qualified: while some studies report variability due to lack of inbred strains, the Ldlr-/- hamster model has shown highly consistent and human-like lesions.
Response 44: Changes have been made on page 11 lines 436 to 438 to address this comment.
Comments 45: The conclusion would benefit from explicitly referring back to earlier discussions on dietary components (cholesterol, cholate, casein, choline). A short sentence synthesizing how diet interacts with genetic background to determine lesion severity would give the conclusion more coherence.
Response 45: Done as suggested (lines 443 to 447 on page 11). Thank you.
Comments 46: The statement that “a combination of two or more rodent models… may be needed” is strong but would be more impactful if paired with one concrete example (e.g., pairing Apoe-/- mice for early lesion formation with hamsters for human-like lipoprotein profiles).
Response 46: Done as suggested.
Comments 47: The discussion of sex hormones and aging is appropriate. You may also note that although young rodents are commonly used, age-related atherosclerosis (e.g., in Ldlr-/- mice) is increasingly studied for translational relevance—acknowledging this trend demonstrates awareness of diverse research aims.
Response 47: We greatly appreciated the very insightful comments. A detailed discussion on the effect of aging on atherosclerosis is felt to be beyond the scope of the discussion of the present review article. It could be more appropriate to address this important issue in depth in a separate paper.
Round 2
Reviewer 2 Report
Comments and Suggestions for Authors
Thank you for your attention.